# Inhibition of CDK1 by RO-3306 Exhibits Anti-Tumorigenic Effects in Ovarian Cancer Cells and a Transgenic Mouse Model of Ovarian Cancer

**DOI:** 10.3390/ijms241512375

**Published:** 2023-08-03

**Authors:** Yu Huang, Yali Fan, Ziyi Zhao, Xin Zhang, Katherine Tucker, Allison Staley, Hongyan Suo, Wenchuan Sun, Xiaochang Shen, Boer Deng, Stuart R. Pierce, Lindsay West, Yajie Yin, Michael J. Emanuele, Chunxiao Zhou, Victoria Bae-Jump

**Affiliations:** 1Department of Gynecologic Oncology, Chongqing University Cancer Hospital, Chongqing 400044, China; 13996274038@163.com; 2Division of Gynecologic Oncology, University of North Carolina at Chapel Hill, Chapel Hill, NC 27599, USA; yaliy@email.unc.edu (Y.F.); zhaoziyi@email.unc.edu (Z.Z.); xixi825@email.unc.edu (X.Z.); katherine.tucker2@unchealth.unc.edu (K.T.); allison.staley@unchealth.unc.edu (A.S.); hongyan@email.unc.edu (H.S.); wenchaun_sun@med.unc.edu (W.S.); xcshen96@email.unc.edu (X.S.); brdeng27@email.unc.edu (B.D.); stuart.pierce@unchealth.unc.edu (S.R.P.); lindsay.west@unchealth.unc.edu (L.W.); yajie_yin@med.unc.edu (Y.Y.); 3Department of Gynecologic Oncology, Beijing Obstetrics and Gynecology Hospital, Capital Medical University, Beijing Maternal and Child Health Care Hospital, Beijing 100054, China; 4Department of Pharmacology, University of North Carolina at Chapel Hill, Chapel Hill, NC 27599, USA; emanuele@email.unc.edu; 5Lineberger Comprehensive Cancer Center, University of North Carolina at Chapel Hill, Chapel Hill, NC 27599, USA

**Keywords:** CDK1, RO-3306, ovarian cancer, apoptosis, invasion

## Abstract

Ovarian cancer is the deadliest gynecological malignancy of the reproductive organs in the United States. Cyclin-dependent kinase 1 (CDK1) is an important cell cycle regulatory protein that specifically controls the G2/M phase transition of the cell cycle. RO-3306 is a selective, ATP-competitive, and cell-permeable CDK1 inhibitor that shows potent anti-tumor activity in multiple pre-clinical models. In this study, we investigated the effect of CDK1 expression on the prognosis of patients with ovarian cancer and the anti-tumorigenic effect of RO-3306 in both ovarian cancer cell lines and a genetically engineered mouse model of high-grade serous ovarian cancer (KpB model). In 147 patients with epithelial ovarian cancer, the overexpression of CDK1 was significantly associated with poor prognosis compared with a low expression group. RO-3306 significantly inhibited cellular proliferation, induced apoptosis, caused cellular stress, and reduced cell migration. The treatment of KpB mice with RO-3306 for four weeks showed a significant decrease in tumor weight under obese and lean conditions without obvious side effects. Overall, our results demonstrate that the inhibition of CDK1 activity by RO-3306 effectively reduces cell proliferation and tumor growth, providing biological evidence for future clinical trials of CDK1 inhibitors in ovarian cancer.

## 1. Introduction

Ovarian cancer (OC) has the highest mortality among gynecologic malignancies and is the fifth leading cause of overall cancer death in women in the United States, estimated to contribute 2.08% of new cancer diagnoses and 4.61% of cancer-related fatalities in women in 2023 [1]. Due to atypical/vague symptoms and a lack of effective diagnostic methods or screening tests in the early stages, more than 70% of patients with ovarian cancer are not diagnosed until the disease has progressed to an advanced stage [2,3]. The conventional treatment for OC patients is tumor debulking surgery in combination with platinum-taxane-based chemotherapy [4]. Although many women achieve remission with standard treatment, the recurrence of cancer and resistance to chemo-therapeutics pose significant challenges for the majority of OC patients [4,5]. Thus, novel therapies, including targeted therapy, are in high demand for the treatment of OC, especially for those patients who have relapsed [6,7].

Cyclin-dependent kinases (CDKs) are a group of proline-directed serine/threonine protein kinases that regulate the cell cycle process by phosphorylating several proteins involved in diverse aspects of cell cycle control [8]. CDK1 is a stimulator of cell cycle progression and specifically controls the G2/M phase transition of the cell cycle and chromosome segregation by phosphorylating many key targets involved in nuclear envelope breakdown and progression via mitosis [9]. The plasticity of CDK1 enables it to adequately compensate for the genetic loss of interphase CDKs, thus making CDK1 the only non-redundant cell cycle regulator in the CDK family [10]. In addition to the control of the cell cycle, CDK1 participates in many biologic processes involved in the growth of cancer cells, including DNA homologous recombination (HR) repair, the maintenance of cancer stem cell properties, mitochondrial bioenergetics, and tumor resistance [8,11,12]. The overexpression of CDK1 has been implicated in the intrinsic modulation of altered Wnt, mTOR, and p53/Rb signaling pathways, suggesting that CDK1 is an essential driver of tumorigenesis [13]. Multi-dimensional bioinformatics analysis from The Cancer Genome Atlas (TCGA) and Gene Expression Omnibus (GEO) datasets confirmed that CDK1 was significantly overexpressed in cancer cells and strongly associated with prognosis in many types of cancer, including OC [13,14,15,16,17]. The inhibition of CDK1 expression by small inhibitors, or CDK1 knockdown, reduces cell viability, induces apoptosis, and impairs cell migration in multiple types of cancer cell lines and xenograft mouse models [18,19,20,21,22,23]. Dinaciclib, a small molecule inhibitor of CDK1, CDK2, CDK5, and CDK9, demonstrated well-tolerated and promising efficacy in phase II and III clinical trials in relapsed multiple myeloma and chronic lymphocytic leukemia but showed low clinical efficacy in solid tumors [24,25,26,27,28]. Milciclib, an inhibitor of CDK1, CDK2, CDK4, and CDK5, exhibited clinical activity in combination with gemcitabine in a phase I clinical trial in refractory solid tumors [29]. Collectively, the evidence supports that targeting CDK1 with small molecule inhibitors deserves further exploration of their therapeutic value for tumors in pre-clinical models and clinical trials [11].

Due to their significant degree of similarity, most inhibitors targeting CDK1 also inactivate CDK2, which is activated throughout the S and G2 phases of the cell cycle. The exception to this is RO-3306, a selective, ATP-competitive, and cell-permeable CDK1 inhibitor that demonstrates potent anti-tumor activity in multiple pre-clinical models [30,31,32]. Given its high selectivity and specificity for CDK1, this current study used RO-3306 to address the anti-tumor potential of CDK1 inhibition in OC cell lines and a transgenic mouse model of OC. The effects of CDK1 inhibition by RO-3306 on cell viability, apoptosis, cell cycle distribution, and invasion were analyzed in OC cells. The effects of RO-3306 on tumor growth in a genetically engineered mouse model of OC (KpB mouse model) under obese and lean conditions were also analyzed. This current study finds that targeting CDK1 by RO-3306 shows anti-proliferative and anti-metastatic effects in OC cells and the KpB mouse model.

## 2. Results

### 2.1. Effect of CDK1 on Prognosis in Patients with OC

To explore the prognostic values of CDK1 expression in OC, we first examined the expression of CDK1 protein by IHC staining in 147 OC tissues. The median patient age was 51.43 years old, with the youngest patient at 30 years and the oldest patient at 76 years. Of the 147 patients in our group, 72 cases were serous type; and 25, 24, and 26 cases were mucinous, endometrioid, and clear types, respectively. Detailed patient clinical information is given in Table 1. All OC samples were stained with CDK1, p53, and Ki-67 antibodies (Figure 1A). Among the OC samples, the expression of CDK1 was significantly associated with the FIGO stage, but there was no relationship between other clinical factors, including lymph node metastasis and histological tumor grade. Given the roles of p53 and Ki-67 in carcinogenesis and progression in OC, we further compared the relationship between the expression of CDK1 and p53 and Ki-67. The results showed that the expression of p53 and Ki-67 was positively correlated with the expression of CDK1 in OC tissues. In addition, among 147 patients with OC, 79.59% had high expressions of CDK1, and only 20.41% had low CDK1 expressions. Kaplan–Meier analysis confirmed that the overexpression of Ki-67 and p53 was associated with prognosis, and high expressions of CDK1 were significantly associated with poorer survival compared with the low expression group (Figure 1B, *p* < 0.05). These results suggest that CDK1 is an independent prognostic factor of overall survival in patients with OC.

### 2.2. Effect of RO-3306 on Cell Proliferation in OC Cells

To determine the inhibitory effect of RO-3306 on cell proliferation, five OC cell lines (SKOV3, HEY, PA-1, OVCAR5, and IGROV1) were treated with different concentrations of RO-3306 for 72 h. Cell proliferation was assessed using an MTT assay. RO-3306 significantly inhibited cell proliferation in a dose-dependent manner in all of the OC cell lines. The IC50 values of SKOV3, HEY, PA-1, OVCAR5, and IGROV1 were 16.92, 10.15, 7.24, 8.74, and 13.89 µM, respectively, as shown in Figure 2A. Next, we selected the OVCAR5 and SKOV3 cell lines to detect further effects of RO-3306 on OC cell growth. Given that colony formation assay is the gold standard for examining the effects of cytotoxic agents on cell growth in cancer cells, the SKOV3 and OVCAR5 cells were treated with RO-3306 at 0.5, 5, and 25 µM doses for 48 h, and both cell lines were cultured for an additional 12 days. Consistent with the MTT assay, RO-3306 effectively reduced colony formation in a dose-dependent manner in both cell lines. Colony formation in the OVCAR5 and SKOV3 cells was reduced by 66.12% and 52.5%, respectively, after treatment with 25 µM RO-3306 (Figure 2B). Because RO-3306 is a specific CDK1 inhibitor, Western blotting was used to detect the expression of CDK1 in both cell lines after treatment with RO-3306 for 24 h. As shown in Figure 2C, different concentrations of RO-3306 significantly reduced the expression of CDK1 in a dose-dependent manner in both cell lines. These results suggest that RO-3306 inhibited cell proliferation in OC cells.

### 2.3. Effect of RO-3306 on Cell Cycle Progression in OC Cells

To investigate the underlying mechanism by which RO-3306 inhibited cell growth, the cell cycle profile was analyzed by Cellometer. The OVCAR5 and SKOV3 cells were exposed to different concentrations of RO-3306 for 36 h. As illustrated in Figure 3A, RO-3306 treatment resulted in a dose-dependent reduction in the G1 cell cycle phase and increased the G2 phase in the OVCAR5 cells. The treatment of SKOV3 cells with 5 or 25 µM RO-3306 also significantly reduced the fraction of cells in the G1 phase and increased the proportion of cells in the G2 phase. The population of the G1 phase of OVCAR5 cells and SKOV3 cells reduced from 55.54% to 22.04% and 57.94% to 29.20%, respectively, after 25 µm RO-3306 treatment compared with control groups. Furthermore, 25 µM RO-3306 significantly increased accumulation cells in the G2 phase by 35.75% and 34.55% in OVCAR5 and SKOV3 cells, respectively. Western immunoblotting results showed that RO-3306 increased the expression of p21 and p27, which are both strong inhibitors of cell cycle progression (Figure 3B). These results confirmed that RO-3306 effectively decreased the cell cycle G1 phase in OVCAR5 and SKOV3 cells.

### 2.4. Effect of RO-3306 on Apoptosis in OC Cells

To examine whether RO-3306 induced apoptosis in OC cells, the OVCAR5 and SKOV3 cells were treated with 0.5, 5, and 25 µM RO-3306 for 16 h, followed by Annexin V staining. RO-3306 increased the percentage of Annexin V-positive apoptotic cells in a dose-dependent manner in both cell lines. As shown in Figure 4A, treatment with 25 µM RO-3306 efficiently increased the population of apoptotic cells by 18.01% in OVCAR5 and 23.44% in SKOV3, respectively, compared with the untreated cells. To explore the effect of RO-3306 on mitochondrial apoptosis pathways, cleaved caspase 3, 8, and 9 activities were detected using ELISA assays in both cell lines. After 14 h of treatment, RO-3306 enhanced cleaved caspase 3, 8, and 9 activities in a dose-dependent manner in both cell lines. 25 µM RO-3306 treatment increased cleaved caspase 3, 8, and 9 by 1.29, 0.33, and 0.27 times, respectively, in OVCAR5 cells and by 1.16, 0.29, and 0.35 times, respectively, in SKOV3 cells, compared with the control cells (Figure 4B–D). Western blotting analysis showed that RO-3306 reduced the expression of the anti-apoptotic proteins Mcl-1 and Bcl-2 and induced the pro-apoptotic protein Bax and cleaved PARP expression in both cell lines (Figure 4E). These results suggest RO-3306 induces apoptosis through extrinsic and intrinsic mitochondrial apoptosis pathways in OC cells.

### 2.5. Effect of RO-3306 on Cellular Stress in OC Cells

ROS has the potential to cause cellular damage and is known to be a sensitive indicator of oxidative stress in cancer cells. We examined the ROS level and mitochondrial potential using the DCFH-DA, JC-1, and TMRE assays. As shown in Figure 5A, RO-3306 irritated the production of ROS in a dose-dependent manner in both cell lines after 12 h of treatment. Treatment with 25 µM RO-3306 increased ROS production by 55.63% and 127.61% in OVCAR5 and SKOV3, respectively, compared to the control cells. Treatment of both cell lines with different concentrations of RO-3306 also significantly decreased the level of JC-1 and TMRE in a dose-dependent manner in both cell lines, suggesting that RO-3306 may reduce mitochondrial membrane potential in OC cells. Western blotting results showed that RO-3306 induced the expression of cell stress-related proteins such as PERK, BiP, Calnexin, ATF4, PDI, and CHOP in the OVCAR5 and SKOV3 cell lines after 24 h of treatment (Figure 5B). These results demonstrate that cellular stress may also involve the inhibitory effect of RO-3306 on cell proliferation in OC cells.

### 2.6. Effect of RO-3306 on Tumor Growth in the KpB Mouse Model of OC

Given that RO-3306 had an anti-proliferative effect in vitro and obesity is known to increase the risk for ovarian cancer, we determined that we had to detect the effect of RO-3306 on tumor growth in a transgenic mouse model of OC (KpB mouse model) under obese and lean conditions. The KpB mice were fed with HFD or LFD at three weeks of age and were then divided into four groups (15 mice/per group), including HFD (obese), HFD+RO-3306, LFD (lean), and LFD+RO-3306. When ovarian tumors reached approximately 0.1 × 0.1 cm by palpation, KpB mice were treated with RO-3306 (4 mg/kg, IP, every three days) for four weeks. The initial mean body weight in obese mice at the start of treatment with RO-3306 was 45.6 gm compared to 28.4 gm in lean mice (*p* < 0.01). During the RO-3306 or placebo treatment period, the mice maintained normal activity and normal body weight, and no deaths were observed. Compared to the lean mice, the KpB mice fed an HFD had a significant increase in tumor volume and tumor weight, suggesting that an obese environment favors ovarian cancer growth. This is consistent with our prior works [33,34,35]. The obese and lean KpB mice treated with RO-3306 had significantly reduced tumor volume and tumor weight compared to the placebo. RO-3306 reduced tumor weight by 71.25% in obese mice and by 73.76% in lean mice, respectively, compared to their controls (Figure 6A,B). There was no statistical difference in tumor inhibition rates between obese and lean mice after RO3306 treatment, indicating that RO3306 exhibited antitumor activity regardless of obese or lean conditions. We next detected the protein expression of Ki-67, Bcl-xL, and BiP in ovarian tumors after RO-3306 treatment by IHC under obese and lean conditions (Figure 6C). The expression of Ki-67 was significantly reduced by 50.8% and 35.9% in the obese and lean KpB mice treated with RO-3306, respectively, compared to the control mice. RO-3306 reduced Bcl-xL expression in the treated obese mice by 31.44% and by 18.25% in lean mice and increased the expression of Bip from 29.87% to 53.25% in the tumors from the obese group and from 35.56% to 47.13% in the tumors from the lean mice group. These results suggest that RO-3306 inhibits the growth of OC tumors via cellular stress and apoptotic pathways in vivo.

### 2.7. Effect of RO-3306 on Adhesion and Invasion In Vitro and In Vivo

Given that increasing the activity of CDK1 has been found to enhance the ability of tumor cells to invade and migrate, as shown in other cancer types, we investigated the effect of RO-3306 on adhesion and invasion in OC cell lines and tumors [36,37,38]. The laminin adhesion assay was employed to analyze the effects of RO-3306 on cell adhesion after being treated with 0.5, 5, and 25 µM of this drug for 1.5 to 2 h. Figure 7A shows that the cell adhesion was decreased by 18.14% and 20.62% in OVCAR5 and SKOV3 cells, respectively, after treatment with 25 µM RO-3306. Wound healing assays were used to investigate the capacity of cell migration. Cell migration was significantly inhibited by 5 µM and 25 µM RO-3306 in both cell lines after 48 h of treatment (Figure 7B). Western blotting results showed that different concentrations of RO-3306 significantly reduced the expression of Slug, Snail, β-Catenin, Vimentin, and VEGF-C in both cell lines after 24 h of treatment (Figure 7C). In addition, serum VEGF levels were reduced by 29.51% in RO-3306-treated obese mice and 20.47% in lean mice compared with the placebo control groups (Figure 7D). IHC results also demonstrated that RO-3306 effectively reduced the expression of VEGF in the ovarian tumors of both obese and lean mice in comparison to placebo-treated mice (Figure 7E). Collectively, these results indicate that the effect of RO-3306 on cell invasion may be dependent on the EMT pathway and angiogenesis in OC.

## 3. Discussion

Previous studies have found that the overexpression of CDK1 in OC tissues conferred a significantly worse prognosis in patients with epithelial OC, and the knockdown of CDK1 resulted in the inhibition of cell proliferation, induction of apoptosis, and distribution of cell cycle phase in OC cells [23,39,40]. By analyzing the genome-wide expression profile of human OC tissues, CDK1 has been identified as one of the most important prognostic factors in OC [16]. In this study, we also demonstrated that the overexpression of CDK1 is associated with poor overall survival in patients with epithelial OC as well as positively correlates with the expression of p53 and Ki-67 in OC tissues. Targeting CDK1 by RO-3306 significantly inhibited cell proliferation, induced apoptosis and cellular stress, and reduced cell invasion in OVCAR5 and SKOV3 cell lines. Importantly, treatment with RO-3306 for four weeks resulted in significant inhibition of tumor growth in the KpB OC mouse model under obese and lean conditions, with a concomitant decrease in the expression of Ki-67, VEGF, and Bcl-xL, and an increase in the expression of BiP. Overall, these results suggest that CDK1 is an ideal target for the treatment of OC.

Emerging evidence supports the important biologic role of CDK1 in controlling the cell cycle in many cancers, and the complexes of CDK1 with cyclin A and cyclin B are sufficient to drive cancer cell cycle progression via the phosphorylation of hundreds of proteins at multiple sites [13,41]. The inhibition of CDK1 usually induces cell cycle arrest at the G2/M phase and results in the induction of cell apoptosis [32]. Although the mechanism by which CDK1 inhibition induces mitochondria-initiated apoptosis is not well understood, several studies have demonstrated that the CDK1/cyclin B complex regulates apoptosis via the stabilization of the mitochondria, modifying several pro-apoptosis proteins and anti-apoptotic proteins, and mediating FOXO1 activity [12,42]. Prior studies have found that targeting CDK1 by RO-3306 induced the expression of cleaved caspase 3, cleaved-PARP, and BAD in a dose- and time-dependent manner in the OVCA-429 and OVCAR-3 OC cell lines [39]. We found that RO-3306-induced apoptosis was also associated with both cleaved caspase 3, 8, and 9 activations and decreased the expression of Bcl-2 and Mcl-1 in a dose-dependent manner in the OVCAR5 and SKOV3 cells. Additionally, the expression of p21 and p27 was upregulated in both OC cell lines after treatment with RO-3306 for 24 h. Given that p21 and p27 are checkpoint CDK inhibitors and have both oncosuppressive and oncopromoting properties, we propose that RO-3306 induces mitochondria-initiated apoptosis that is dependent on cell cycle regulation [43,44,45].

The CDK1/cyclin B complex effectively regulates mitochondrial dynamics, mitochondrial protein influx, and bioenergetics via the phosphorylation of mitochondrial and Bcl family proteins to sustain metabolic reprogramming in the face of increased energy requirements for cancer cell survival [12,46]. Several anti-tumor agents, such as dinaciclib, apigenin, LX1570, and matrine, were found to induce cellular stress via CDK1 inhibition in cancer cells [46,47,48,49]. Our results indicate that the inhibition of CDK1 by RO-3306 significantly increases cellular ROS production and decreases mitochondrial membrane potential, accompanied by changes in cellular stress-related proteins in ovarian cancer cells. Given that RO-3306 reduced the expression of Bcl-2 and Mcl-1, the mechanism by which RO-3306 induces cellular stress may depend on the process of phosphorylation of Bcl family proteins by CDK1 in OC cells.

OC has a unique metastatic pattern in which ovarian cancer cells disseminate directly from the primary tumor site, spread easily into the peritoneal cavity, and invade peritoneal organs, a process regulated by EMT and the integrin-mediated upregulation of matrix metalloproteinases (MMPs) [50,51]. Increased CDK1 activity has been linked to an increased ability for cancer cells to invade and migrate [37,38]. Meanwhile, the knockdown of CDK1 significantly reduced the ability of invasion and migration via the regulation of EMT proteins in adrenocortical carcinoma cells and head and neck squamous cell carcinomas (HNSCCs) cells [52,53]. Similar results were recently found in OC cells, where the knockdown of CDK1 reduced cell proliferation, migration, and invasion in CAOV3 cells [54]. Our results reveal that RO-3306 significantly inhibited cell adhesion, migration, and invasion in a dose-dependent manner in the OVCAR5 and SKOV3 cells. The results of Western blotting confirm that RO-3306 downregulates the expression of EMT-related proteins in OC cells, suggesting that CDK1 is a driver that regulates adhesion and invasion by modulating the EMT process in OC cells. Additionally, we found that RO-3306 reduces the expression of VEGF in OC cells and OC tissues. Given that VEGF promotes the EMT process and CDK1 enhances VEGF mRNA expression in cancer cells [55,56,57], we speculate that the inhibition of VEGF by CDK1 may be another pathway to inhibit the invasion of OC.

The results of the expression of CDK1 in OC patients show that the expression of CDK1 is upregulated in OC, especially in advanced-stage patients. More importantly, CDK1 upregulation shows a significant positive correlation with the expression of Ki-67 and mutant p53. The high expression of Ki-67 has been associated with poor overall survival in OC patients, whereas p53 mutations are the most frequent genetic alternation in OC and a genomic hallmark in advanced/high-grade serous (HGS) OCs [58,59,60,61]. Overall, these data further support the biological role of CDK1 in tumorigenesis and progression in OC. Since we analyzed only 147 ovarian cancer cases in this study due to a lack of follow-up information for some patients, there may be potential bias in the selection of these patients due to sample size. We are continuing to assess the prognostic value of CDK1, p53, and Ki-67 expression in combination for OC outcomes as part of our ongoing work.

In conclusion, CDK1 expression and function are clearly implicated in the carcinogenesis and prognosis of OC. Targeting CDK1 by RO-3306 inhibited cell proliferation, induced cellular stress, and apoptosis, reduced invasive capacity, and reduced tumor growth in OC cells and a transgenic mouse model of OC. Although these results were generated in two ovarian cancer cell lines and a single ovarian cancer transgenic mouse model, our data provide biological evidence for future clinical trials of CDK1 inhibitors in OC.

## 4. Materials and Methods

### 4.1. Patients and OC Samples

A total of 147 patients with epithelial OC admitted to Chongqing University Cancer Hospital were registered in this study between 2017 and 2021. The protocol for this study was approved by the Ethics Committee of Chongqing University Cancer Hospital. Written informed consent was obtained from each patient. Experienced pathologists in the Department of Pathology of Chongqing University Cancer Hospital established pathological diagnoses on H&E staining slides based on World Health Organization (WHO) standards. The International Federation of Gynecology and Obstetrics (FIGO) staging system was used to determine a patient’s clinical stage. The clinical and pathological characteristics of the patients are shown in Table 1. Immunohistochemistry (IHC) staining for CDK1, p53, and Ki67 was performed at the Department of Pathology of Chongqing University Cancer Hospital.

### 4.2. Cell Culture and Reagents

Five OC cell lines, PA-1, Hey, IGROV1, OVCAR5, and SKOV3, were used in this study. OVCAR5, Hey and IGROV1 were cultured in RPMI supplemented with 5 or 10% fetal bovine serum (FBS). PA-1 cells were cultured in DMEM medium with 10% FBS, and SKOV3 cells were maintained in McCoy’s 5A medium with 10% FBS. All media were supplemented with 2 mM L-glutamine, 100 U/mL penicillin, and 100 µg/mL streptomycin. Cells were cultured in an incubator under 5% CO_2_ at 37 °C. RO-3306 was purchased from Sigma (Cat# SML0569, St. Louis, MO, USA). All antibodies were purchased from Cell Signaling Technology (Beverly, MA, USA) and Abclonal Science (Woburn, MA, USA). All other chemicals were purchased from Thermo Fisher Scientific (Waltham, MA, USA), Sigma, and MedChemExpress (Monmouth Junction, NJ, USA).

### 4.3. Cell Proliferation Assay

The OVCAR5 and SKOV3 cells were cultured overnight in 96-well plates with a concentration of 4000 cells/well. The cells were then treated with various concentrations of RO-3306 for 72 h. A total of 5 µL of 3-(4,5-Dimethyl-2-thiazolyl)-2,5-diphenyl-2H-tetrazolium Bromide (MTT, 5 mg/mL) were added into each well, followed by incubation for 1 to 1.5 h. MTT absorbance was detected using a microplate reader (Tecan, Morrisville, NC, USA) at a wavelength of 562 nm after mixing 100 µL dimethyl sulfoxide (DMSO) into each well. The effect of RO-3306 on cell proliferation was calculated as a percentage of control, and the IC50 was calculated using the AAT Bioquest IC50 calculator (Pleasanton, CA, USA). Each experiment was performed at least three times for consistency.

### 4.4. Colony Assays

The OVCAR5 and SKOV3 cells were plated in 6-well plates at the density of 400 cells/well. After 24 h of growth, the cells were treated with 0.5, 5, and 5 µM of RO-3306 or vehicle for 48 h. The cells were continuously cultured for 12 days with fresh medium changes every three or four days. Cells were stabilized with methanol and stained with 0.5% crystal violet buffer for 30 min. The clones were imaged and quantified using Image J software (V1.8.0) (National Institutes of Health, Bethesda, MD, USA).

### 4.5. Cell Cycle Assay

The OVCAR5 and SKOV3 cells were grown in 6-well plates overnight and then treated with different concentrations of RO-3306 for 36 h. Cells were harvested with trypsin, and the cell pellets were resuspended in ice-cold 90% methanol for 30 min. The methanol was removed, and the cells were resuspended in 100 µL RNase A solution for 30 min followed by incubation with propidium iodide (2 mg/mL) at room temperature in dark conditions for 20 min. The profile of cell cycle progression was measured using Cellometer (Nexcelom, Lawrence, MA, USA). FCS4 express software (Molecular Devices, Sunnyvale, CA, USA) was used to analyze the distribution of the cell cycle. The experiment was repeated at least three times for consistency.

### 4.6. Annexin V Assay

The OVCAR5 and SKOV3 cells were treated with various concentrations of RO-3306 for 16 h in 6-well plates. Cells were harvested with 0.25% trypsin and rinsed three times in cold PBS. Cells were then suspended in 100 µL binding buffer containing Annexin V-FITC and 0.5 µL of propidium iodide for 15 min. The proportion of Annexin V-labeled cells was quantified using Cellometer, and the results were analyzed via FCS4 express software. The experiments were repeated three times for consistency.

### 4.7. Cleaved Caspase 3, 8, and 9 ELISA Assays

The OVCAR5 and SKOV3 cells were seeded in 6-well plates at a concentration of 2.5 × 10^5^ cells/well for 24 h and then treated with different concentrations of RO-3306 for 14 h. After washing each well with cold PBS, 150–180 µL of 1× caspase lysis buffer was added to each well, and the concentration of protein in each well was detected using a BCA kit (Thermo Fisher). An equal amount of protein from each well and reaction buffer with caspase 3, 8, and 9 substrates were added to a new black 96-well plate and incubated at 37 °C for 15 min. The fluorescence intensities for cleaved caspase 3 (Ex/Em = 400/505), cleaved caspase 8 (Ex/Em = 376/482), and cleaved caspase 9 (Ex/Em = 341/441) were recorded using a Tecan microplate reader. These assays were repeated three times for consistency.

### 4.8. Reactive Oxygen Species (ROS) Assay

Moreover, 5.0 × 10^3^ cells/well of OVCAR5 and SKOV3 cells were cultured in 96-well plates overnight, followed by exposure to varying doses of RO-3306 for 12 h. Then, 15 µM DCFH-DA was added into each well, and the plate was incubated at 37 °C for 30 min. The ROS production was determined by detecting the fluorescence intensity at Ex/Em = 485/530 nm using a Tecan microplate reader. All experiments were performed at least three times for consistency.

### 4.9. Mitochondrial Membrane Potential Assays

Mitochondrial membrane potential was analyzed with specific fluorescent probes for JC-1 and TMRE (AAT Bioquest, Sunnyvale, CA, USA). Both cells were seeded in 96-well plates overnight and then treated with different concentrations of RO-3306 for 12 h. Cells were then treated with 2 µM JC-1 or 800 µM TMRE for 30 min at 37 °C. The levels of fluorescent probes were measured using a Tecan plate reader. The wavelength of fluorescence intensity for JC-1 was Ex/Em = 480/590 nm (red) and Ex/Em = 535/590 nm (green) and was Ex/Em = 549/575 nm for the TMRE assay. Each experiment was repeated three times for consistency.

### 4.10. Adhesion Assay

Additionally, 96-well plates were pre-coated with 100 µL laminin-1 (10 µg/mL) and incubated at 37 °C for one hour. The plate was rinsed two times with cold PBS, then 200 µL of blocking buffer was added into each well for 45 to 60 min at 37 °C. 1.5 × 10^4^ cells of OVCAR5 and SKOV3, and varying concentrations of RO-3306 were added into each well, followed by incubation at 37 °C for 1.5 to two hours. The plate was washed with PBS to remove nonadherent cells, then 100 µL of 5% glutaraldehyde was added to fix cells for 30 min. A total of 100 µL of 0.1% crystal violet was added to each well for 30 min. The cells were then washed repeatedly with water, and 100 µL of 10% acetic acid was added to each well to solubilize the dye. The absorbance was measured at a wavelength of 562 nm using a Tecan microplate reader. Each experiment was repeated three times for consistency.

### 4.11. Wound Healing Assay

The OVCAR5 and SKOV3 cells were plated at 4.0 × 10^5^ cells/well in 6-well plates and cultured overnight. A 200 µL pipette tip was used to draw straight lines in each well, and the cells were treated with 0.5, 5, and 25 µM RO-3306 for 48 h. Photos were taken at 24 and 48 h after scratching, and the width of the wound was measured with Image J software (National Institutes of Health, Bethesda, MD, USA). The experiment was repeated three times for consistency.

### 4.12. Western Immunoblotting

The OVCAR5 and SKOV3 cells were treated with varying concentrations of RO-3306 for six to 36 h, and then the total protein was harvested with radioimmunoprecipitation assay buffer (RIPA buffer, Thermo Fisher). The protein concentration was measured via BCA assay. Equal amounts of protein were electrophoresed using 10–12% acrylamide gel and transferred onto a PVDF membrane. The membranes were incubated with primary antibody (1:1000) overnight at 4 °C and then washed three times with tris-buffered saline-tween 20 (TBS-T) buffer. The appropriate secondary antibody was incubated with membranes for one hour at room temperature. Proteins were visualized using SuperSignal West Pico Substrate (Thermo Scientific) via the ChemiDoc image system (Bio Rad, Hercules, CA, USA). Each experiment was repeated at least twice for consistency.

### 4.13. KpB Transgenic Mouse Model of OC

The KpB (K18-gT^+/−^ 121; p53^fl/fl^; Brca1^fl/fl^) transgenic mouse model of high-grade serous ovarian cancer was used in this study as previously described [62,63]. The animal protocol was approved by the University of North Carolina at Chapel Hill Institutional Animal Care and Use Committee (IACUC, protocol #21-229). Sixty KpB mice were housed on a 12 h light, 12 h dark cycle, with free access to food and water. Given that obesity is a risk factor for the development of ovarian cancer potentially via CDK signaling, we evaluated the efficacy of RO-3306 in both obese and lean mice [64]. To imitate diet-induced obesity (DIO), one-half of the KpB female mice were fed a high-fat diet (HFD, 60% calories from fat), and the other one-half were fed a low-fat diet (LFD, 10% calories from fat, Research Diets) at three weeks of age. At six to eight weeks old, all mice were injected with 5 µL recombinant adenovirus Ad5- CMV-Cre (2.5 × 10^10^ P.F.U, Transfer Vector Core, University of Iowa) into the left ovarian bursa cavity. The mice kept feeding with HFD and LFD chow and were examined weekly for the appearance of ovarian tumors via abdominal palpation. Once tumors had reached an average size of 0.1 × 0.1 cm in diameter by palpation, the mice were assigned into four groups: HFD control, LFD control, HFD+RO-3306, and LFD+RO-3306 (N = 15 mice/group). KpB mice were treated for four weeks with RO-3306 (4 mg/kg, IP, every three days) when the ovarian tumor reached approximately 0.1 × 0.1 cm. All animals were checked daily for any signs of toxicity and weighed weekly during the treatment. Tumor size was monitored using palpation twice a week until the tumor grew to a size suitable for caliper measurements. All mice were sacrificed via CO_2_ asphyxiation after four weeks of treatment. Ovarian tumors and serum were collected. One-half of the tumor tissues were frozen at −80 °C, and the remaining tissues and serum were embedded in paraffin blocks. Blood serum was also stored at −80 °C. Ovarian tumor volumes were calculated as follows: (width2 × length)/2.

### 4.14. Serum VEGF Assay

VEGF productions of mice serum were detected via VEGF ELISA Kit (#MMV00, R&D Systems, Minneapolis, MN, USA). Each sample from the HFD control, LFD control, HFD+RO-3306, and LFD+RO-3306 groups was measured in duplicate. VEGF levels were measured at 570 nm using a Tecan plate reader.

### 4.15. IHC for Ovarian Tumors of KpB Mice

The ovarian tumor slides (4 μm) were incubated in protein block solution (Dako, Santa Clara, CA, USA) for one hour and then primary antibodies (VEGF, Ki67, Bcl-xL, and BiP) were probed overnight at 4 °C. The slides were then washed with TBS-T and incubated with secondary antibodies at room temperature for one hour. Each slide was further processed using ABC-Staining Kits (Vector Labs, Burlingame, CA, USA) and hematoxylin. All slides were scanned via Motic (Feasterville, PA, USA) and analyzed using ImagePro software V2.1.8 (Vista, CA, USA). The degree of staining was scored according to the percentage of positive tumor cells: 0 (negative and <5% cells), 1 (5–25%), 2 (25–50%), 3 (50–75%), and 4 (>75%). The immunostaining intensity of each tumor slide was evaluated as 0 (negative), 1 (weak), 2 (moderate), and 3 (strong). Scores less than or equal to 3 were considered the low expression of CDK1; scores greater than 3 were considered strong positive CDK1 overexpression when the staining score and intensity score were added together [40].

### 4.16. Statistical Analysis

All data were reported as mean ± SD from three independent assays. Both Student’s t-tests and one-way ANOVA tests were used in this study. The Kaplan–Meier analysis was used to estimate survival rates. GraphPad Prism 8 (La Jolla, CA, USA) statistical software was employed to calculate the comparisons. All tests were two-sided with *p* < 0.05 considered significant.

## Figures and Tables

**Figure 1 ijms-24-12375-f001:**
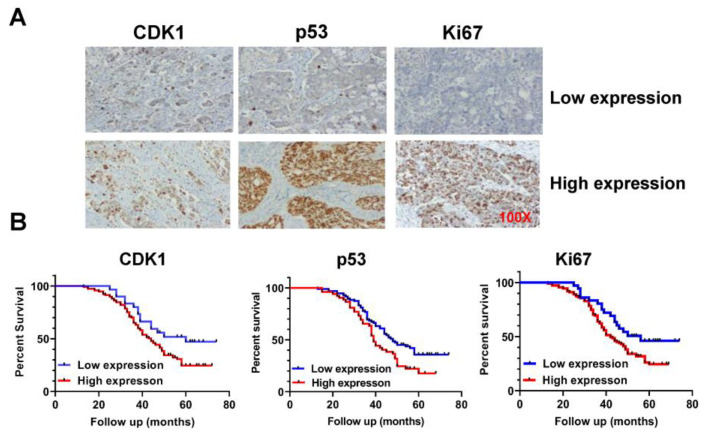
Overexpression of CDK1 was associated with poor prognosis in patients with OC. A total of 147 epithelial OC tissues were stained with antibodies to CDK1, p53, and Ki-67 via IHC (**A**). Kaplan–Meier survival curves show that CDK1 overexpression is significantly associated with poorer survival compared with OC patients with low CDK1 expression (**B**) *p* < 0.01.

**Figure 2 ijms-24-12375-f002:**
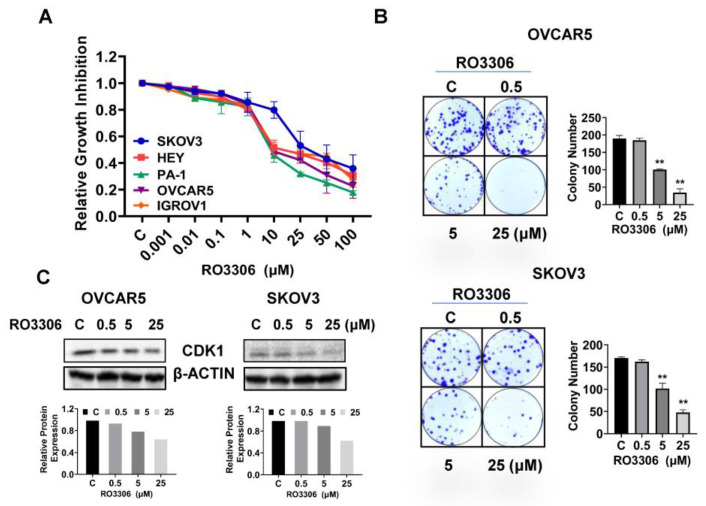
RO-3306 inhibited cell proliferation in OC cells. The Hey, PA-1, IGROV1, OVCAR5, and SKOV3 cells were treated with different concentrations of RO-3306 for 72 h. MTT assay showed that RO-3306 inhibited cell proliferation in a dose-dependent manner in both cell lines (**A**). Colony assays showed that RO-3306 reduced colony formation in both cell lines after treating cells with RO-3306 for 48 h and continuing to culture cells for 12 days (**B**). Western blotting confirmed that RO-3306 decreased the expression of CDK1 in both cell lines after 24 h of treatment (**C**). ** *p* < 0.01.

**Figure 3 ijms-24-12375-f003:**
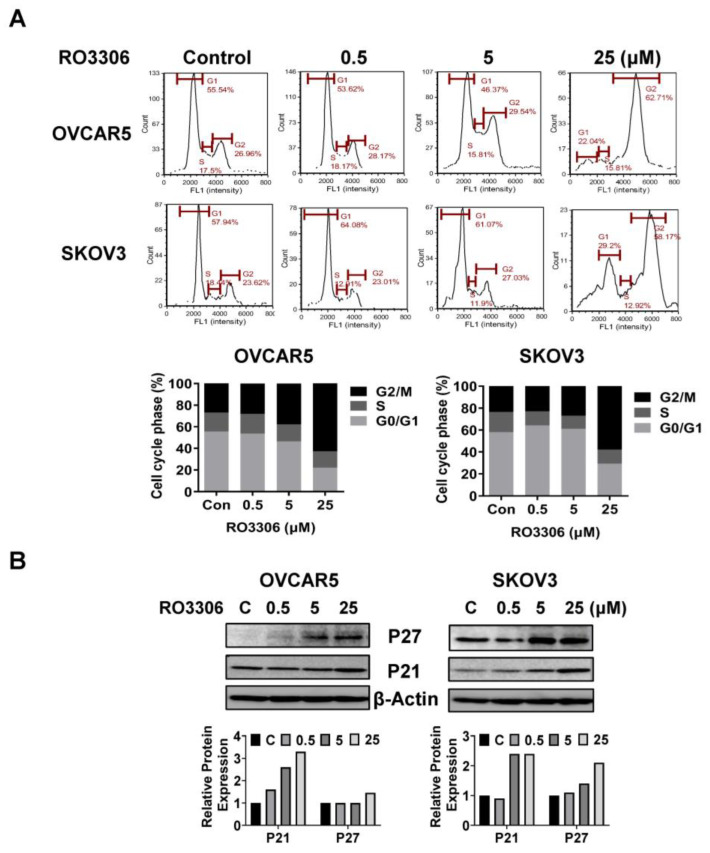
RO-3306 caused cell cycle G2 phase arrest in OC cells. Treatment with different concentrations of RO-3306 reduced the cell cycle G1 phase and caused cell cycle G2 arrest in the OVCAR5 and SKOV3 cell lines. Representative results are shown in (**A**), showing the distribution of the cell cycle following 36 h of treatment. Histograms showed the results of the cell cycle distribution after RO-3306 treatment. Western blotting analysis demonstrated changes in p21 and p27 expression for both cell lines after 24 h of treatment (**B**). All the experiments performed were repeated at least three times.

**Figure 4 ijms-24-12375-f004:**
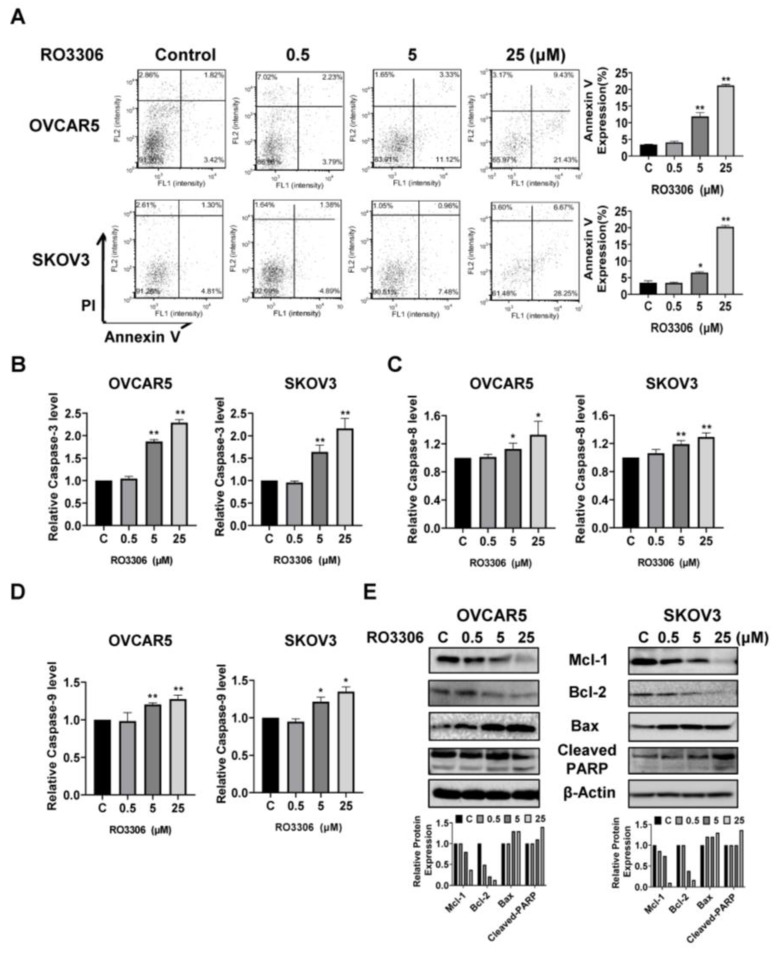
RO-3306 induced apoptosis in OC cells. The OVCAR5 and SKOV3 cells were treated with RO-3306 for 16 h, followed by Annexin V staining. RO-3306 increased the expression of Annexin V in both cell lines (**A**). ELISA assays were used to detect cleaved caspase 3, 8, and 9 activities after treatment with RO-3306 for 14 h. The results showed that RO-3306 (5 or 25 µM) enhanced cleaved caspase 3, 8, and 9 activities (**B**–**D**). Treatment of both cell lines with RO-3306 for 24 h increased the expression of Bax and cleaved PARP and decreased MCL-1 and Bcl-2 expression (**E**). All the experiments were repeated at least three times. * *p* < 0.05, ** *p* < 0.01.

**Figure 5 ijms-24-12375-f005:**
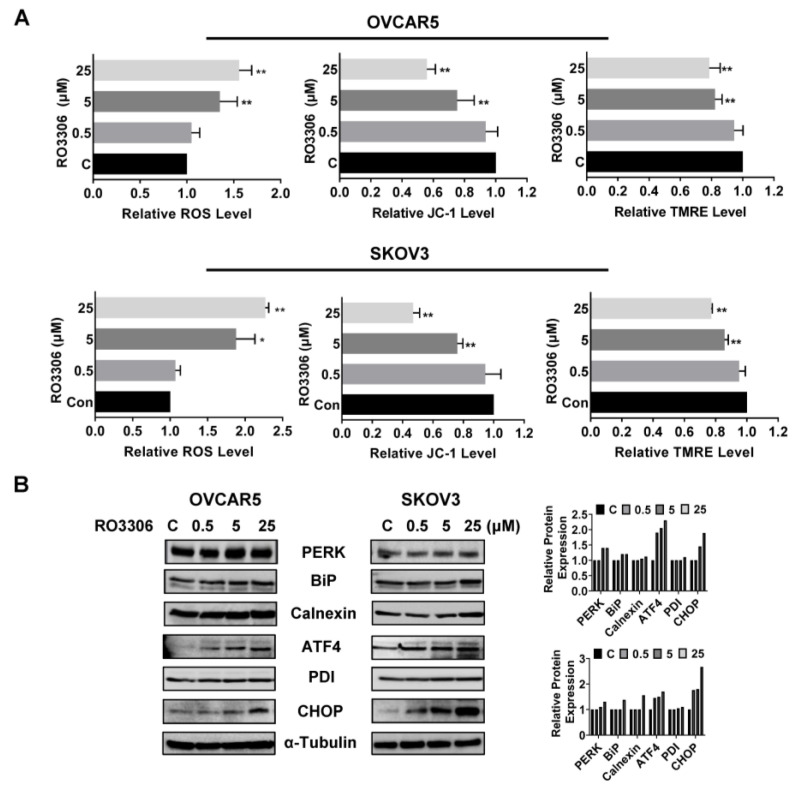
RO-3306 induced cellular stress in OC cells. The OVCAR5 and SKOV3 cells were treated with RO-3306 (0.5, 5 and 25 µM) for 12 h. Intracellular ROS productions were measured using DCFH-DA assays. JC1 and TMRE assays were used to examine mitochondrial membrane potential. RO-3306 significantly increased cellular ROS production and reduced mitochondrial membrane potential in both cell lines (**A**). Western blotting analysis showed that treatment with RO-3306 increased the expression of cellular stress-related proteins such as PERK, BiP, ATF4, CHOP, Calnexin, and PDI in both cell lines (**B**). * *p* < 0.05, ** *p* < 0.01.

**Figure 6 ijms-24-12375-f006:**
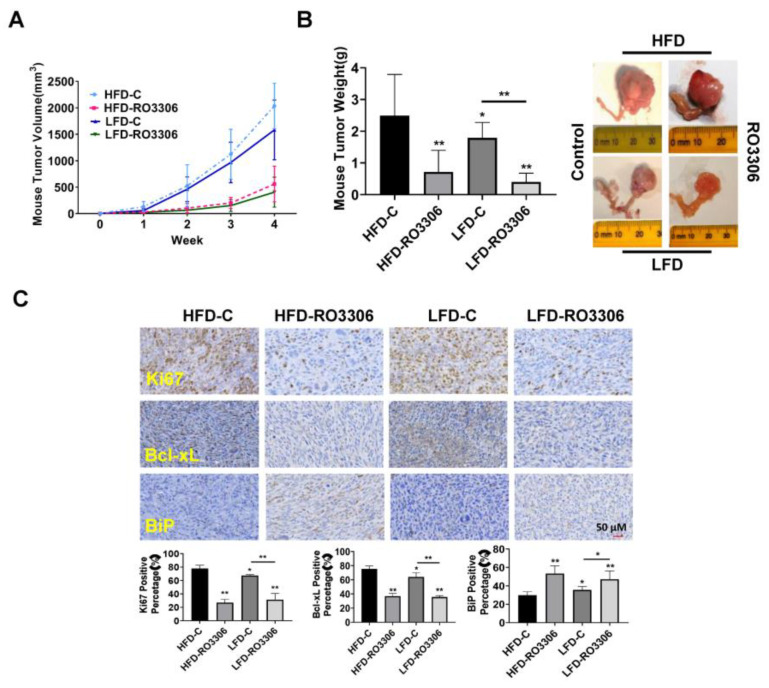
RO-3306 inhibited tumor growth in a transgenic mouse model of OC. The KpB mice were fed with HFD (obese) or LFD (lean) at three weeks of age. The obese or lean KpB mice received RO-3306 (4 mg/kg, IP, every three days) for four weeks after the ovarian tumor reached approximately 0.1 × 0.1 cm in size. RO-3306 treatment significantly decreased tumor volume and tumor weight in obese and lean KpB mice (**A**,**B**). Immunohistochemical analysis of ovarian tumors from the RO-3306 and control groups showed downregulation of Ki-67 and Bcl-xL expression and upregulation of BiP expression under obese and lean conditions (**C**). * *p* < 0.05, ** *p* < 0.01.

**Figure 7 ijms-24-12375-f007:**
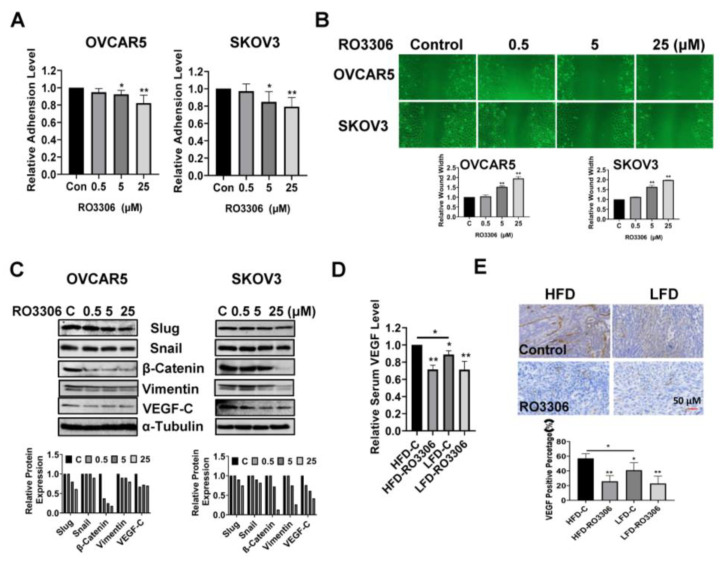
RO-3306 reduced cell migration and angiogenesis in OC cells and KpB mice. Treatment of OVCAR5 and SKOV3 cells with RO-3306 for 1.5 to 2 h significantly reduced cell adhesion (**A**). Wound healing assays showed that RO-3306 inhibited cell migratory capacity after 48 h of treatment in both cell lines (**B**). Western blotting results indicated that RO-3306 decreased the expression of EMT-related proteins, including Slug, Snail, β-Catenin, Vimentin, and VEGF-C in the OVCAR5 and SKOV3 cells (**C**). Serum VEGF production was decreased by RO-3306 treatment in KpB mice (**D**). Immunohistochemical analysis confirmed that RO-3306 reduced the expression of VEGF in ovarian tissues in obese and lean KpB mice (**E**). * *p* < 0.05, ** *p* < 0.01.

**Table 1 ijms-24-12375-t001:** Clinical characteristics of 147 cases of ovarian cancer.

Parameters		Number of Patients	CDK1 Expression		*p* Value	X^2^
			Low (≤3)	High (>3)		
**Age**	≥50	67	19	48	<0.05	3.933
	<50	80	11	69		
**Pathological Type**	Mucinous	25	8	17	0.32	3.477
	Endometrioid	24	5	19		
	Clear Cell	26	6	20		
	Serous	72	11	61		
**FIGO Stage**	I	25	12	13	<0.01	51.346
	II	34	4	30		
	III	68	1	67		
	IV	20	13	7		
**BMI**	≤24	91	14	77	>0.05	2.943
	>24	56	16	40		
**Metastasis**	Yes	88	14	74	>0.05	2.085
	No	59	16	43		
**p53** **Expression**	Low (≤3)	53	20	33	<0.01	13.698
	High (>3)	94	10	84		
**Ki67** **Expression**	Low (≤3)	53	20	33	<0.01	26.656
	High (>3)	94	10	84		

## Data Availability

The data that support the findings of this study are available from the corresponding authors upon reasonable request.

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
