# Peer review of "Inhibition of CDK1 by RO-3306 Exhibits Anti-Tumorigenic Effects in Ovarian Cancer Cells and a Transgenic Mouse Model of Ovarian Cancer"

_ijms, 2023, doi:10.3390/ijms241512375_

Round 1

Reviewer 1 Report

Evidence suggests that CDK1 is a promising target for OC treatment. The study's findings could potentially pave the way for future clinical trials involving CDK1 inhibitors as a novel therapeutic strategy for ovarian cancer patients.The text appears to be well-written and provides a comprehensive analysis of the study's findings. The introduction effectively sets the stage, the methods are sound, and the results are presented clearly. The discussion offers valuable insights into the implications of the research and potential areas for further investigation.

I suggest expanding the sample size in the patient cohort and including a larger number of ovarian cancer samples.

I suggest expanding the sample size in the patient cohort and including a larger number of ovarian cancer samples, validating with independent cohorts, and adding control groups for non-OC patients histologies.  

Author Response

.The authors are very grateful to the reviewers for their comments and suggestions. Increasing the sample size will undoubtedly increase the reliability of CDK1's prognostic effect in ovarian cancer in this study. In the original design of this study, we planned to recruit 250 to 300 patients with ovarian cancer. Since our hospital is located in Southwest China, which is an underdeveloped area, it is difficult for us to obtain follow-up information by phones or emails for some patients. Although we performed IHC staining on more than 200 cases, in the end we only collected complete information from 147 ovarian cancer patients.

At the beginning of this study, we also planned to use 17 cases of borderline ovarian tumors in the same period as a control group. However, compared with epithelial ovarian cancer, CDK1 expression was lower in which 5 cases were negative for CDK1. We also cannot get follow-up data in another 6 cases. Thus, we are unable to use Kaplan-Meier to calculate survival curves in borderline ovarian tumors to compare the effect of CDK1 on the prognosis of epithelial ovarian cancer.

Reviewer 2 Report

The authors of this manuscript "Inhibition of CDK1 by RO-3306 exhibits anti-tumorigenic and fects in ovarian cancer cells and a transgenic mouse model of ovarian cancer", in my opinion have chosen an efficient target (cdk1) to test an inhibitor in the treatment of ovarian cancer, one of the most impacting cancers for women's health. From my point of view, this research was conducted in an impeccable manner, in fact the authors first deal with the correlation study of the expression of cdk1 and tumor aggressiveness markers such as p53 and ki67 on tissues from cancer patients. Then they perform studies on cells and mice, the cell lines chosen and the methods applied both in vitro and in vivo are suitable for this research. The introduction, findings and discussion are comprehensive and the references are correct. I consider this manuscript complete and publishable

Author Response

We thank reviewer for the comments

Round 2

Reviewer 1 Report

The study presents compelling in vitro results, but it would be valuable to include in vivo data from the KpB transgenic mouse model to further validate the potential therapeutic effects of RO-3306. Adding data on tumor growth inhibition, survival rates, and potential adverse effects in the mouse model would strengthen the translational relevance of the study.

The authors are encouraged to provide a comprehensive acknowledgment of the limitations inherent in their study. It would be prudent to address potential biases in patient selection, the scope and applicability of in vitro assays, as well as the use of a single mouse model.

Author Response

The study presents compelling in vitro results, but it would be valuable to include in vivo data from the KpB transgenic mouse model to further validate the potential therapeutic effects of RO-3306. Adding data on tumor growth inhibition, survival rates, and potential adverse effects in the mouse model would strengthen the translational relevance of the study.

Answer: Thanks for the comments and suggestions. We have added data on tumor growth inhibition and adverse effects to the manuscript. In addition, we did not perform survival experiments with RO3306 on KpB mice due to UNC animal policy.

The authors are encouraged to provide a comprehensive acknowledgment of the limitations inherent in their study. It would be prudent to address potential biases in patient selection, the scope and applicability of in vitro assays, as well as the use of a single mouse model。

Answer: This is a great point. We have added the information to the Discussion.